# Left ventricular long-axis function in hypertrophic cardiomyopathy - Relationships between e', early diastolic excursion and duration, and systolic excursion

**Roger E. Peverill●\*, Bon Chou, Lesley Donelan, Wai-ee Thai**

Monash Cardiovascular Research Centre, MonashHeart and Department of Medicine, School of Clinical Sciences at Monash Health, Monash University and Monash Health, Clayton, Victoria, Australia

\* roger.peverill@monash.edu

## Abstract

### Background

The peak velocity of early diastolic mitral annular motion (e') is believed to provide sensitive detection of left ventricular (LV) diastolic dysfunction in hypertrophic cardiomyopathy (HCM), but other aspects of LV long-axis function in HCM have received less attention. Systolic mitral annular excursion (SExc) is also reduced in HCM and must be an intrinsic limitation to the extent of the subsequent motion during diastole. However, the effects of HCM on excursion during early diastole (EDExc) and atrial contraction (AExc), the duration of early diastolic motion (EDDur), and the relationships of EDExc with SExc, and of e' with EDExc and EDDur, are all unknown.

### Methods

The study group was 22 subjects with HCM and there were 22 age and sex matched control subjects. SExc, EDExc, e', AExc and EDDur were measured from pulsed wave tissue Doppler signals acquired from the septal and lateral walls. In the combined group of HCM and control subjects, multivariate analyses were performed to identify independent predictors of EDExc and e' for both LV walls.

### Results

SExc, EDExc and e' were all lower, and EDDur was longer in the HCM group compared to the control group for both LV walls (p<0.05 for all). In contrast, AExc was lower for the septal wall in the HCM group (p<0.05), but not different between the groups for the lateral wall. In regression analyses of the combined group, EDExc was positively correlated with SExc, and SExc explained 57–86% of the variances in septal and lateral EDExc, e' was positively correlated with EDExc, and EDExc explained 58–68% of the variances of e', whereas the combination of EDExc with EDDur explained 87–92% of the variances in e'. A diagnosis of HCM was not an independent predictor of EDExc when in combination with SExc, but was a minor contributor to the prediction of e' in combination with EDExc and EDDur.

**Data Availability Statement:** All relevant data are within the manuscript and its Supporting Information files.

**Funding:** The author(s) received no specific funding for this work.

**Competing interests:** The authors have declared that no competing interests exist.

**Abbreviations:** a', peak velocity of atrial contraction mediated motion of the mitral annulus; AExc, atrial contraction mediated excursion of the mitral annulus; BP, blood pressure; BSA, body surface area; e', peak velocity of early diastolic mitral annular motion; EDAT, duration of acceleration of early diastolic mitral annular motion; EDDT, duration of deceleration of early diastolic mitral annular motion; EDDur, total duration of early diastolic mitral annular motion; EDExc, early diastolic excursion of the mitral annulus; HCM, hypertrophic cardiomyopathy; IVRT, isovolumic relaxation time; LV, left ventricular; s', peak velocity of systolic mitral annular motion; SDur, duration of systolic mitral annular motion; SExc, systolic excursion of the mitral annulus; TDI, tissue Doppler imaging.

## Conclusion

In HCM, the decrease in LV longitudinal contraction is the major mechanism accounting for a lower EDExc, the lower e'is accounted for by contributions from the lower EDExc and prolongation of early diastolic motion, and there is no atrial compensation for the reduction of long-axis contraction.

## Introduction

Hypertrophic cardiomyopathy (HCM) is a heritable heart disease, in which the classical phenotype is of myocardial hypertrophy occurring independently of an increase in afterload [1]. HCM has been traditionally thought of as a disease of diastolic rather than systolic dysfunction, with evidence for the absence of systolic dysfunction being the presence of a normal or even supranormal left ventricular (LV) ejection fraction (EF) [2]. However, that the LVEF in HCM is out of keeping with other measurements of LV systolic function became apparent once long-axis LV function became the subject of study. A lower systolic peak velocity of mitral annular motion (s') using tissue Doppler imaging (TDI) in HCM subjects with the classical phenotype was first described in 2001 [3], and then confirmed in subsequent studies [4–6]. Other abnormalities of systolic long-axis function reported in HCM include reduced mitral annular systolic excursion (SExc) [7] and prolonged systolic time intervals [5].

With respect to diastolic abnormalities, echocardiographic evidence of abnormal LV filling in HCM was first reported in the 1970s based on measurements of wall motion [8, 9], and then confirmed in the 1980s with Doppler findings, which included prolongation of the isovolumic relaxation and deceleration times, and decrease in the peak velocity of early diastolic transmitral flow (E) [10]. Following the introduction of TDI, reduction in the LV long-axis peak early diastolic velocity (e') was found to be a sensitive test for the detection of LV dysfunction in HCM [3–5]. A reduction in e'was found in some studies of genetically confirmed HCM even prior to the development of hypertrophy [3, 4, 11]. However, while e'has been reported to be inversely correlated with the time constant of relaxation derived from LV pressure measurement [12], and thus assumed to be a marker of diastolic function, there is increasing evidence that e'is affected by factors other than the speed of LV relaxation. Thus, e'has also been reported to be correlated with s'[13, 14] and the extent of long-axis contraction [15], and changes in e'during inotropic interventions appear to be determined in large part by changes in the extent of the preceding contraction [16, 17].

Peak TDI velocities have been one of the mainstays of the assessment of LV long-axis function, but additional value from also considering excursion of the mitral annulus during systole (SExc), early diastole (EDExc) and atrial contraction (AExc), in conjunction with systolic and diastolic TDI time intervals, has been demonstrated recently in a study investigating the determinants of the aging-related reduction in e'[18]. In this study, SExc, EDExc and e'all decreased with age, the main determinant of EDExc was SExc, the main determinant of e'was EDExc, and there were no consistent changes in either LV long-axis systolic duration (SDur) or early diastolic duration (EDDur) with increasing age. In contrast to SExc and EDExc, AExc increased with age, but to a lesser extent than EDExc decreased with age. HCM is also accompanied by reductions in both SExc [7] and e'[3–5], and a positive correlation between SExc and e'has been described [15], but the effects of HCM on EDExc and AExc have not been investigated, and the relationships of EDExc with SExc, and of e'with EDExc, are unknown. Furthermore, a similar relationship between e'and EDExc to that seen during aging cannot be

assumed, particularly if HCM is also accompanied by a change in EDDur. Accordingly, the aims of this study were to compare the above TDI variables in subjects with HCM and age and sex matched control subjects, to investigate the extent to which EDExc is determined by SExc, and e'is determined by EDExc and EDDur, and also to investigate whether these independent variables account for the expected differences in EDExc and e'between HCM and control subjects.

## Methods

### Subjects

The study design was approved by the Monash Health Human Research and Ethics committee and all clinical investigation was conducted according to the principles expressed in the Declaration of Helsinki. Subjects with HCM who had undergone echocardiography were identified retrospectively as fitting the criteria of a clinical diagnosis of hypertrophic cardiomyopathy, no history of coronary artery disease, hypertension or diabetes, a normal LV ejection fraction (LVEF; >50%) and no abnormality of regional LV contraction. Subjects with HCM had been advised to withhold beta blocker medication for >48 hours prior to their echocardiogram. A group of healthy subjects were identified to be a control group from subjects who had undergone echocardiography in our laboratory, found to have a normal echocardiogram, and who were matched for age and sex with the HCM subjects. Inclusion criteria for healthy control subjects were no history of cardiac disease, diabetes or hypertension, no cardiac medication, and a systolic BP <160 and a diastolic BP <90 mm Hg at the time of the study. All control subjects also needed to have echocardiographic findings of a LVEF >50%, no regional wall motion abnormality and no more than mild valvular disease. The need to obtain consent from the study subjects was waived and analysis was performed on anonymized data. Height and weight were measured immediately prior to the echocardiographic study and body surface area (BSA) and body mass index were calculated using standard formulae. Blood pressure (BP) was measured during the echocardiographic study with the subject in a supine position.

### Echocardiography

Echocardiography was performed using a Phillips Sonos 5500 and studies were stored digitally and were measured offline using Xcelera V1.2 L4 SP2 (Philips, Amsterdam, The Netherlands). Transmitral Doppler signals were obtained with the sample volume at the mitral leaflet tips. Pulsed-wave TDI was performed in the apical 4-chamber view and TDI signals of longitudinal mitral annular motion were recorded during non-forced end-expiration apnoea at both septal and lateral borders of the mitral annulus after optimising parallel alignment of the ultrasound beam and positioning of the sample volume [19].

Measurements were made from the systolic TDI signal of s'and the velocity time integral (SExc), from the early diastolic TDI signal of e' and the velocity time integral (EDExc), and from the atrial contraction TDI signal of a'and the velocity time integral (AExc), as previously described [15, 20]. E/septal e', E/lateral e'and E/average e'were calculated, with the understanding that E/e'does not provide an accurate estimation of mean left atrial pressure in HCM subjects [21]. The reported results are averages of 3 consecutive beats. Time intervals were measured from the onset of the QRS complex to the onset and end of the TDI systolic signal, and to the onset, peak and end of the TDI early diastolic signal. An example of a pulsed wave TDI signal obtained from the lateral border of the mitral annulus in a control subject is presented in Fig 1, showing measurements of the velocity time integrals of the systolic, early diastolic and atrial contraction signals, and of the systolic and diastolic time intervals. The time interval between the end of the systolic signal and the commencement of the early diastolic

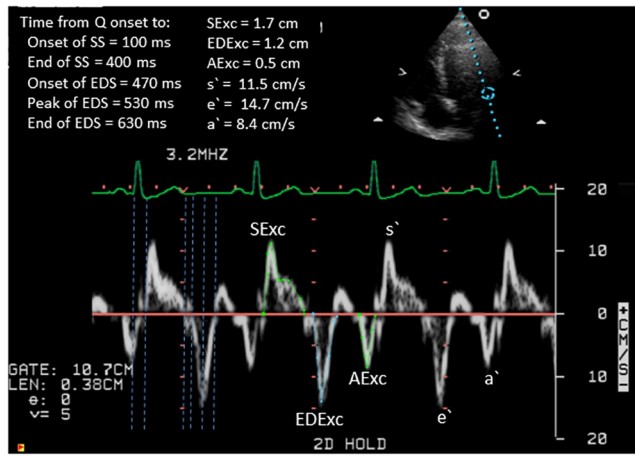

**Fig 1. An example of pulsed wave tissue Doppler imaging from the lateral border of the mitral annulus in a control subject, illustrating the methods used for tracing the velocity time integrals of the systolic, early diastolic and atrial contraction signals and for measuring the systolic and diastolic time intervals.**

signal was calculated as a TDI long-axis equivalent of the isovolumic relaxation time (IVRT) [22]. Also calculated from the TDI time intervals were the systolic duration (SDur), the early diastolic duration (EDDur), and its components of the duration of early diastolic acceleration (EDAT) and the duration of early diastolic deceleration (EDDT). Time intervals were measured for transmitral flow from the onset of the QRS to the onset, peak and end of the E wave, and the early diastolic duration and deceleration times were calculated. The heart rate was calculated from the R-R intervals of the relevant TDI and transmitral signals.

## Statistical analysis

Statistical analysis was performed using Systat V13 (Systat Software, Chicago, IL, USA). Continuous variables are presented as mean ± SD or median [range] if not normally distributed. An unpaired t test or Mann-Whitney test, as appropriate, was used to compare continuous variables in HCM and control subjects. Repeated measures analysis of variance was used to compare equivalent TDI and transmitral time intervals within the HCM and control groups, with the Sidak test used to determine the significance of pairwise comparisons. For univariate linear regression analysis, the r value has been provided, and for multivariate analyses the partial standard correlation coefficient (β) value has been provided. The coefficient of determination has been adjusted for the number of terms in multivariate models (adjusted $r^2$) and used to estimate the variance (as a percentage) of a dependent variable explained by that model. The decisions regarding inclusion of variables in multivariate models were based on the aims of the study. Multivariate linear regression analysis in the combined HCM and control group was performed to assess the relationships between long-axis systolic and diastolic variables. HCM versus control was included as a dummy variable (1 v 0) as a last step in multivariate models to see if it accounted for further variance in the dependent variable after inclusion of the other independent variables. Separate multivariate analyses were also performed for e'in the HCM and control groups. A p value of <0.05 was considered significant.

## Results

The characteristics of the HCM and control subjects are shown in Table 1 and were similar between the groups, except for a trend to a higher body mass index in HCM subjects. All

**Table 1. Comparison of HCM and control subjects.**

|  | HCM | Control | *P* |
|---|---|---|---|
| Male: Female | 16:6 | 16:6 | NS |
| Age (years) | 39±10 | 40±9 | NS |
| Height (cm) | 174±8 | 175±8 | NS |
| Weight (kg) | 79±13 | 75±9 | NS |
| Body surface area (m$^2$) | 1.93±0.22 | 1.91±0.15 | NS |
| Body mass index (kg/m$^2$) | 26.0±3.1 | 24.7±2.1 | 0.086 |
| Blood pressure (mmHg) | 118±14/69±11* | 113±11/72±8 | NS |
| Heart rate | 64±11 | 65±11 | NS |
| Septal wall thickness | 2.0±0.5 | 0.9±0.1 | <0.001 |
| Posterior wall thickness | 1.3±0.4 | 0.9±0.1 | <0.001 |

* BP measurements were not available in 3 subjects.

HCM subjects had asymmetric septal hypertrophy and as a group had a larger septal wall thickness and larger posterior wall thickness compared to control subjects. There were 11 HCM subjects with a resting LV peak outflow tract gradient >50 mmHg, 9 subjects with an inducible gradient (exercise or Valsalva) >30 mmHg, and only 2 HCM subjects did not have a resting or inducible gradient >30 mmHg.

The transmitral Doppler variables and TDI Doppler velocities and excursions for the septal and lateral walls in HCM and control subjects are shown in Table 2, and an example showing

**Table 2. Comparison of transmitral flow variables and left ventricular long-axis excursions and velocities for the septal and lateral walls in HCM and control subjects.**

|  | HCM | Control | *p* |
|---|---|---|---|
| **Transmitral Doppler** |  |  |  |
| E (cm/s) | 82±21 | 79±17 | 0.65 |
| A (cm/s) | 63±21 | 50±12 | 0.018 |
| E/A | 1.5±0.7 | 1.6±0.5 | 0.32 |
| Deceleration time (ms) | 223±80 | 182±25 | 0.03 |
| E/ septal e' | 14.6 [7.8–36.7] | 8.9 [5.8–19.6] | <0.001 |
| E/lateral e' | 10.8 [5.4–44.3] | 5.4 [3.0–12.5] | <0.001 |
| E/average e' | 11.8 [6.4–40.1] | 6.6 [4.2–15.3] | <0.001 |
| **Septal** |  |  |  |
| SExc (cm) | 1.1±0.2 | 1.3±0.3 | 0.001 |
| EDExc (cm) | 0.6±0.2 | 0.8±0.2 | 0.004 |
| AExc (cm) | 0.5±0.1 | 0.6±0.1 | 0.044 |
| s'(cm/s) | 7.2±1.6 | 8.5±1.6 | 0.013 |
| e'(cm/s) | 5.4±1.7 | 9.0±2.0 | <0.001 |
| a'(cm/s) | 7.2±2.0 | 9.4±2.8 | 0.006 |
| **Lateral** |  |  |  |
| SExc (cm) | 1.2±0.4 | 1.7±0.3 | <0.001 |
| EDExc (cm) | 0.8±0.3 | 1.2±0.3 | <0.001 |
| AExc (cm) | 0.5±0.1 | 0.5±0.1 | 0.073 |
| s'(cm/s) | 8.4±2.8 | 12.4±2.8 | <0.001 |
| e'(cm/s) | 7.2±2.9 | 15.4±3.7 | <0.001 |
| a'(cm/s) | 9.3±2.9 | 10.3±2.2 | 0.21 |

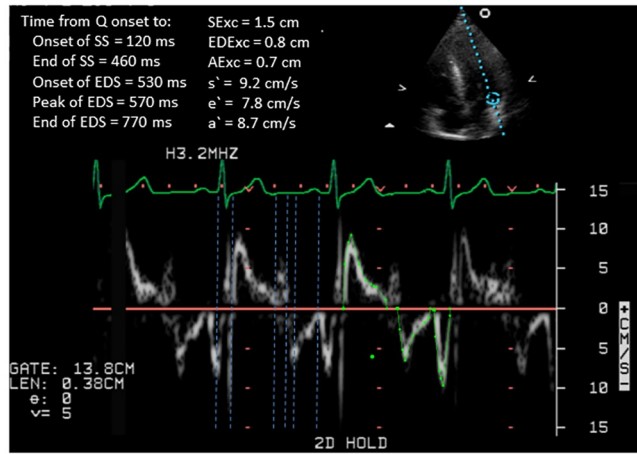

**Fig 2. An example of pulsed wave tissue Doppler imaging from the lateral border of the mitral annulus in a subject with hypertrophic cardiomyopathy, showing the results of systolic and diastolic excursions, peak velocities and time intervals.**

measurements of TDI velocities and excursions from the lateral wall in a subject with HCM is shown in Fig 2. The transmitral A was increased and the deceleration time was prolonged in HCM, but the E and E/A were similar in the two groups. Of the TDI velocities and excursions, the magnitudes of all but lateral AExc and aʻwere lower in the HCM group. Compared to control subjects, in HCM the average value of eʻwas 40% lower for the septal wall and 55% lower for the lateral wall, whereas septal and lateral EDExc were 25% and 33% lower, respectively. The differences between the mean values of SExc and EDExc for the HCM and control subjects were of similar, albeit not identical, magnitude for both septal and lateral walls. E/septal eʻ, E/lateral eʻand E/average eʻwere all larger in HCM subjects.

### Long-axis time intervals for the septal and lateral walls

TDI time intervals for the septal and lateral walls and transmitral time intervals in the HCM and control groups are shown in Table 3 and measurements of TDI time intervals for the lateral wall in a subject with HCM are shown in Fig 2. The heart rates at the time of the measurement of the TDI and transmitral signals did not differ within or between the groups (p>0.5 for all comparisons). The systolic intervals for the lateral wall were all longer in the HCM than the control group, but there were no significant differences between the HCM and control group for the septal wall systolic intervals. All the lateral wall systolic time intervals were delayed by comparison to the septal wall time intervals in the HCM group (p<0.05 for all), but the systolic time intervals were similar for the two walls in the control group.

Most of the diastolic time intervals for both walls were longer in the HCM group, with the exception of lateral EDAT. The average septal EDDur was 28% longer and the average lateral EDDur was 27% longer in HCM compared to control subjects. In the control group, the onset of E was slightly earlier than the onset of the lateral early diastolic TDI signal, which was in turn slightly earlier than the onset of the septal early diagnostic signal (p<0.05 for both comparisons). On the other hand, there was no difference in the time to the peak, or the time to the end, of the early diastolic signals for the septal and lateral walls in the control group, whereas the end of E occurred later than the end of both the septal and lateral early diastolic TDI signals (p<0.05 for both comparisons). Similar to the control group, in the HCM group the onset of E was earlier than the onset of the septal or lateral early diastolic signals (p<0.05

**Table 3. Comparison of left ventricular long-axis time intervals for the septal and lateral walls in HCM and control subjects.**

|  | HCM | Control | *p* |
|---|---|---|---|
| **Septal** | | | |
| Time to onset of systolic signal (ms) | 115±20 | 105±15 | 0.065 |
| Time to end of systolic signal (ms) | 404±46 | 400±26 | 0.69 |
| SDur (ms) | 246±63 | 239±45 | 0.69 |
| Isovolumic relaxation time (ms) | 116±51 | 71±24 | 0.001 |
| Time to onset of early diastolic signal (ms) | 546±67 | 485±32 | 0.001 |
| Time to peak of early diastolic signal (ms) | 612±69 | 542±33 | <0.001 |
| Time to end of early diastolic signal (ms) | 758±98 | 653±41 | <0.001 |
| EDDur (ms) | 213±57 | 167±27 | 0.002 |
| EDAT (ms) | 69±22 | 57±12 | 0.039 |
| EDDT (ms) | 143±54 | 110±23 | 0.013 |
| **Lateral** | | | |
| Time to onset of systolic signal (ms) | 134±38 | 104±23 | 0.003 |
| Time to end of systolic signal (ms) | 427±48 | 400±28 | 0.031 |
| SDur (ms) | 292±49 | 237±49 | 0.001 |
| Isovolumic relaxation time (ms) | 85±38 | 59±18 | 0.008 |
| Time to onset of early diastolic signal (ms) | 545±70 | 474±35 | <0.001 |
| Time to peak of early diastolic signal (ms) | 606±74 | 538±29 | <0.001 |
| Time to end of early diastolic signal (ms) | 753±107 | 638±44 | <0.001 |
| EDDur (ms) | 208±60 | 164±42 | 0.007 |
| EDAT (ms) | 65±19 | 63±16 | 0.82 |
| EDDT (ms) | 144±62 | 100±33 | 0.007 |
| **Transmitral** | | | |
| Time to onset of E (ms) | 497±58 | 457±35 | 0.008 |
| Time to peak of E (ms) | 572±61 | 536±32 | 0.021 |
| Time to end of E (ms) | 766±93 | 697±40 | 0.004 |
| Early diastolic flow duration (ms) | 269±49 | 240±25 | 0.021 |
| Early diastolic flow acceleration time (ms) | 81±20 | 77±17 | 0.44 |
| Early diastolic flow deceleration time (ms) | 223±80 | 182±25 | <0.001 |

for both comparisons), and the times to the peak and end of the early diastolic signal were not different for either wall. However, in contrast to the control group, in the HCM group there was no difference between the onset of the septal and lateral early diastolic signals, and there were no differences between the times to the end of E and to the end of the septal and lateral early diastolic signals.

As an additional check on the mechanisms underlying the changes in septal and lateral wall TDI intervals associated with HCM, multivariate analyses of the time intervals of onset of QRS to end of the systolic signal, SDur, IVRT, onset of QRS to end of the early diastolic signal and EDur were performed, with age, heart rate and diagnosis as independent variables. The results are shown in Table 4, with age not shown as it was not a contributor to any of the models. There were both similarities and differences between the findings for the septal and lateral walls. Heart rate was an inverse correlate of all the time intervals for both walls except SDur and IVRT. A HCM diagnosis was associated with a longer IVRT, a longer time to the end of the diastolic signal and a longer EDur for both walls. A HCM diagnosis was also associated with a longer SDur and a trend to a longer time to the end of the systolic signal for the lateral wall, but had no effect on systolic time intervals for the septal wall.

**Table 4. Univariate regression analysis and multivariate models of systolic and diastolic time intervals in the combined HCM and control groups.**

| Dependent variable | Independent variables | Univariate r | Multivariate β | P value in multivariate model | Cumulative adjusted r² |
|---|---|---|---|---|---|
| **Septal wall** | | | | | |
| QRS to end of systolic signal | Heart rate | -0.64 | -0.64 | <0.001 | 0.38 |
| | HCM | 0.04 | 0.013 | 0.92 | 0.38 |
| SDur | Heart rate | 0.05 | | 0.73 | 0 |
| | HCM | 0.04 | | 0.79 | 0 |
| IVRT | Heart rate | -0.06 | -0.03 | 0.80 | 0 |
| | HCM | 0.50 | 0.50 | 0.001 | 0.21 |
| QRS to end of diastolic signal | Heart rate | -0.53 | -0.51 | <0.001 | 0.26 |
| | HCM | 0.57 | 0.55 | <0.001 | 0.56 |
| EDDur | Heart rate | -0.31 | -0.29 | 0.036 | 0.07 |
| | HCM | 0.45 | 0.44 | 0.002 | 0.25 |
| **Lateral wall** | | | | | |
| QRS to end of systolic signal | Heart rate | -0.48 | -0.49 | 0.001 | 0.21 |
| | HCM | 0.25 | 0.26 | 0.052 | 0.27 |
| SDur | Heart rate | -0.05 | -0.06 | 0.65 | 0 |
| | HCM | 0.45 | 0.45 | 0.002 | 0.17 |
| IVRT | Heart rate | -0.00 | -0.017 | 0.90 | 0 |
| | HCM | 0.41 | 0.41 | 0.007 | 0.12 |
| QRS to end of diastolic signal | Heart rate | -0.45 | -0.47 | <0.001 | 0.19 |
| | HCM | 0.57 | 0.59 | <0.001 | 0.52 |
| EDDur | Heart rate | -0.30 | -0.32 | 0.024 | 0.07 |
| | HCM | 0.39 | 0.40 | 0.005 | 0.22 |

## Determinants of s', EDExc and e'in univariate and multivariate analyses of the combined HCM and control groups

The results of univariate and multivariate linear regression analyses of septal and lateral s', including the respective SExc and the heart rate as independent variables, are shown in Table 5. Only 49–55% of the variances in septal and lateral s'were explained by the respective SExc. Heart rate was a univariate correlate of septal s', was not a univariate correlate of lateral s', but with the addition of heart rate to SExc, there was an increase in the variance explained

**Table 5. Univariate regression findings and multivariate models of septal and lateral s', EDExc and e'in the combined HCM and control groups.**

| Dependent variable | Independent variables | Univariate r | Multivariate β | P value in multivariate model | Cumulative adjusted r² |
|---|---|---|---|---|---|
| Septal s' | Septal SExc | 0.71 | 0.73 | <0.001 | 0.49 |
| | Heart rate | 0.24 | 0.30 | 0.006 | 0.56 |
| Lateral s' | Lateral SExc | 0.75 | 0.79 | <0.001 | 0.55 |
| | Heart rate | 0.08 | 0.22 | 0.031 | 0.59 |
| Septal EDExc | Septal SExc | 0.76 | 0.74 | <0.001 | 0.57 |
| Lateral EDExc | Lateral SExc | 0.93 | 0.93 | <0.001 | 0.86 |
| Septal e' | Septal EDExc | 0.76 | 0.83 | <0.001 | 0.57 |
| | Septal EDDur | -0.45 | -0.54 | <0.001 | 0.87 |
| | HCM | -0.70 | -0.19 | 0.007 | 0.89 |
| Lateral e' | Lateral EDExc | 0.82 | 0.73 | <0.001 | 0.66 |
| | Lateral EDDur | -0.43 | -0.36 | <0.001 | 0.87 |
| | HCM | -0.77 | -0.21 | 0.007 | 0.89 |

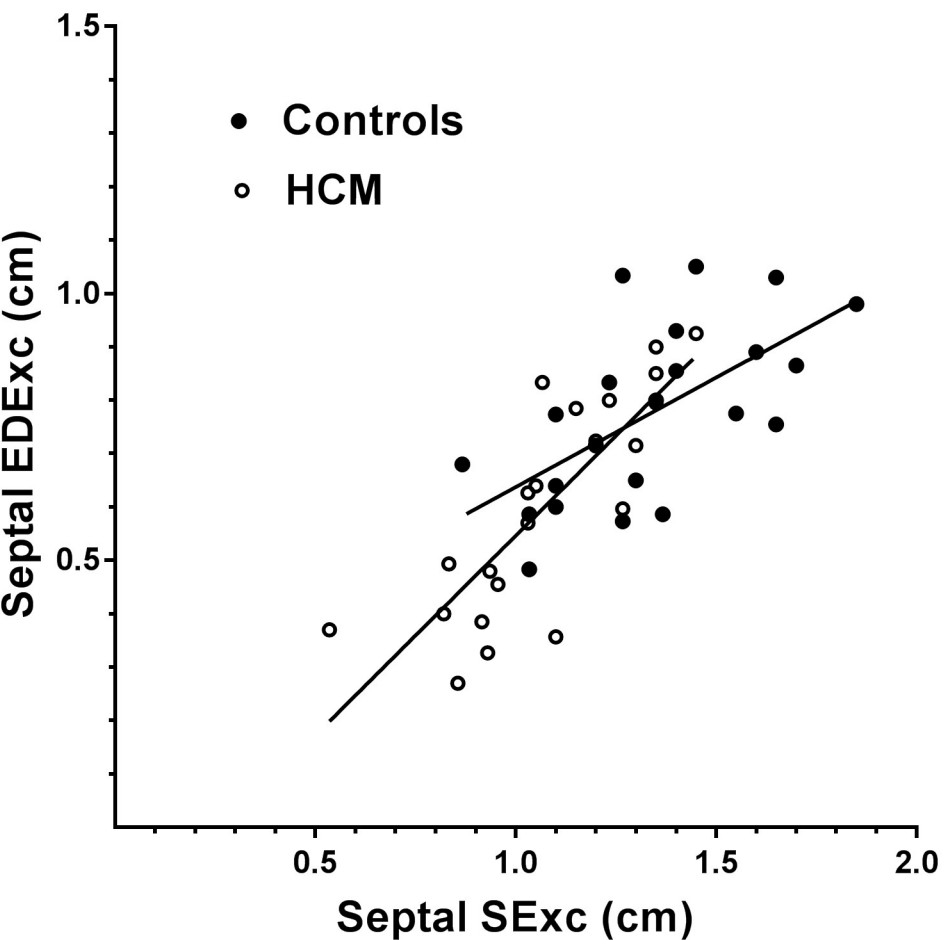

**Fig 3. Scatter plot showing the relationships of EDExc with SExc for the septal wall in the HCM and control groups, with regression lines shown separately for each group.** There was no difference between the slopes of the regression lines for the two groups.

for both septal and lateral s'and there was an increase in the standardized correlation coefficient of heart rate for both walls. A diagnosis of HCM made no contribution to models of s'which included SExc and heart rate (p>0.15 for both).

Scatter plots of the relationships of EDExc with SExc in the combined HCM and control groups for the septal and lateral walls are shown in Figs 3 and 4, with the regression lines for HCM and control subjects shown separately. In the combined group, septal and lateral EDExc were positively correlated with their respective SExc, and SExc accounted for 57–86% of the variances in EDExc (Table 5). A diagnosis of HCM made no contribution to the models of EDExc which included SExc (p>0.5 for both).

Univariate and multivariate linear regression analyses of septal and lateral e', including the respective EDExc and EDDur and HCM diagnosis as independent variables, are shown in Table 4. Septal and lateral e'were both positively correlated with their respective EDExc, and EDExc explained 57–66% of the variances in e'. The addition of EDDur to EDExc further increased the variances explained of both septal and lateral e'to 87% and a minor but significant further improvement to 89% accompanied the addition of a HCM diagnosis to the models.

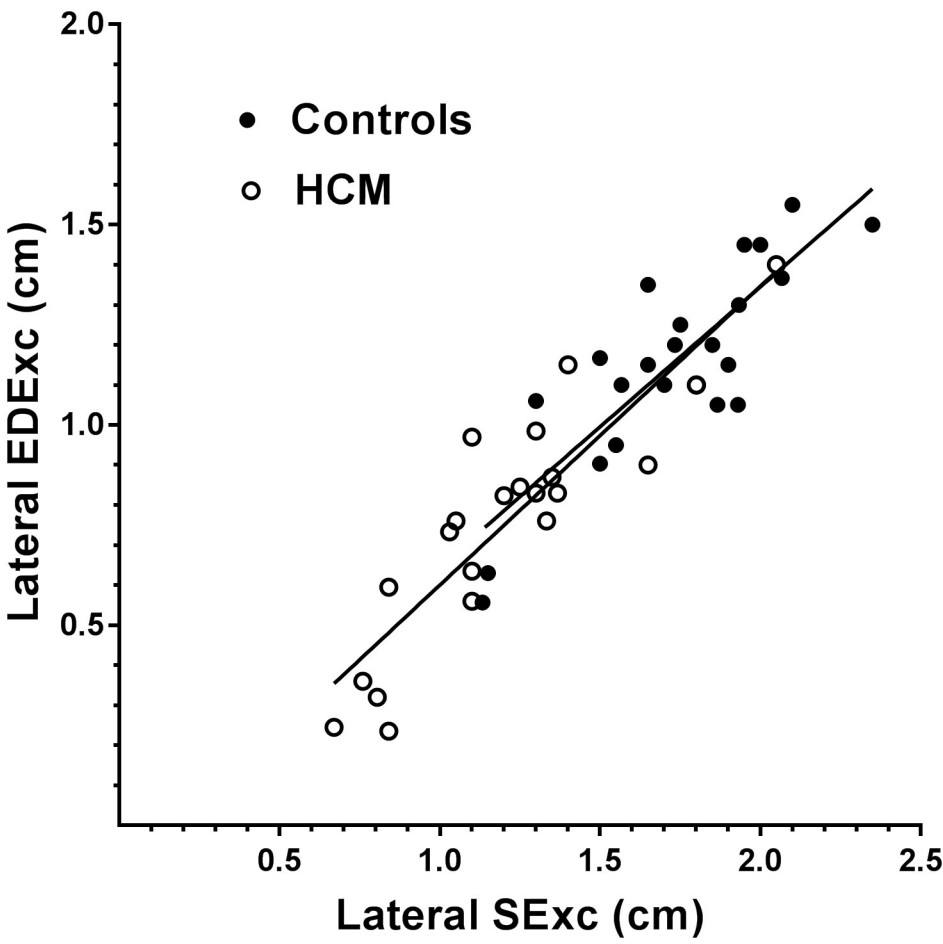

**Fig 4. Scatter plot showing the relationships of EDExc with SExc for the lateral wall in the HCM and control groups, with regression lines shown separately for each group.** There was no difference between the slopes of the regression lines for the two groups.

### Determinants of e'in separate multivariate analyses of the HCM and control groups

Scatter plots of the relationships of e'with EDExc for the septal and lateral walls are shown in Figs 5 and 6, with the regression lines of HCM and control subjects shown separately. Univariate regression analysis and multivariate models of septal and lateral e'in separate analyses of the HCM and control groups are shown in Table 6. Septal and lateral e'were more closely related to EDExc in the control group than in the HCM group, with 63–65% of the variances in e'explained by EDExc in the control group, but only 40–41% explained by EDExc in the HCM group. EDDur made an additional contribution when combined with EDExc to the models of septal and lateral e'in both the control and HCM groups, but EDExc and EDDur each accounted for approximately half of the variances of e'in the HCM group, whereas EDExc made the major contribution to the prediction of e'in the control group.

### Discussion

In this study we have compared TDI measurements of LV long-axis systolic and diastolic excursions, peak velocities and timing in subjects with HCM and age and sex matched control

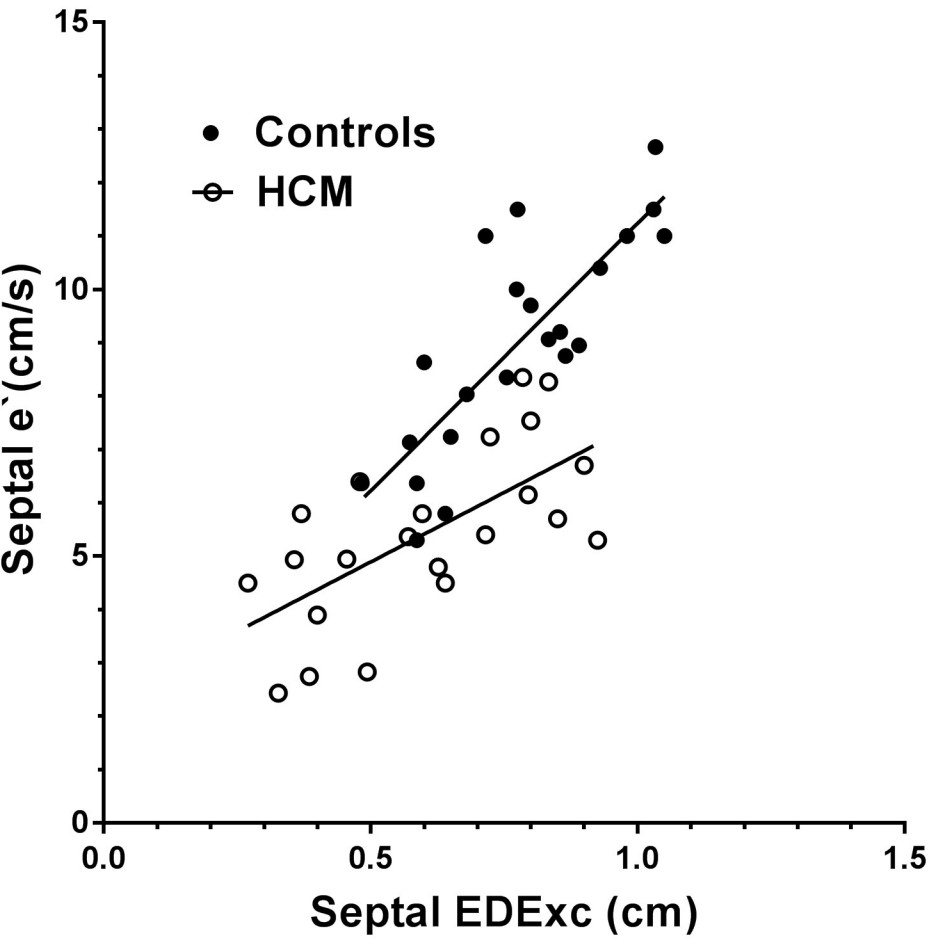

**Fig 5. Scatter plot showing the relationships of e'with EDExc for the septal wall in the HCM and control groups, with regression lines shown separately for each group.** There was a difference between the slopes of the regression lines for the two groups (p = 0.007).

subjects. The main aims were to investigate the extent to which EDExc is determined by SExc, e'is determined by EDExc and EDDur, and whether these independent variables accounted for the expected differences in EDExc and e'between HCM and control subjects. Similar to previous reports on the effects of aging, and consistent with previous reports in HCM, SExc, s'and e'were all lower in HCM compared to control subjects. We have also demonstrated for the first time that EDExc was lower in HCM subjects for both the septal and lateral walls. In contrast to the effects of aging, during which AExc and a'have been found to increase [18, 19], neither AExc nor a'were elevated in HCM, and septal wall AExc and a'were actually both lower in HCM subjects. There was evidence of prolongation of both systolic and diastolic time intervals in HCM compared to control subjects, with some differences evident between the septal and lateral walls. In the combined group of HCM and control subjects, and similar for both walls, SExc explained more than half of the variances of EDExc, and EDExc and EDDur were independent predictors of e'and together explained 87% of the variances in e'. A diagnosis of HCM made no additional contribution to models of EDExc which included SExc, and only made minor contributions in models of e'which included EDExc and EDDur. However, EDDur made a much larger relative contribution to the prediction of e'in HCM than control subjects.

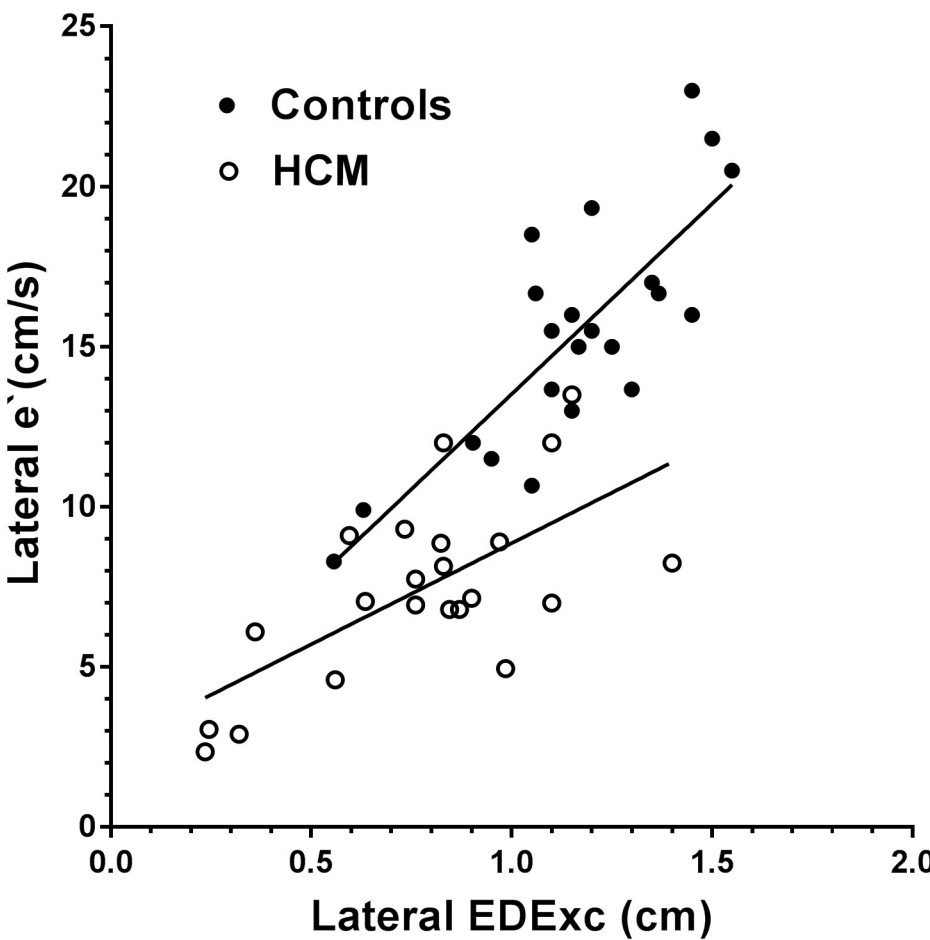

**Fig 6. Scatter plot showing the relationships of e'with EDExc for the lateral wall in the HCM and control groups, with regression lines shown separately for each group.** There was a difference between the slopes of the regression lines for the two groups (p = 0.007).

A fundamental principle of LV long-axis function is that mitral annular excursion during diastole cannot be greater than the excursion which occurred during the preceding systole. Assuming beat-to-beat stability and the absence of any non-standard timing of annular motion (e.g. post ejection shortening), SExc will be of similar magnitude to the sum of EDExc

**Table 6. Univariate regression and multivariate models of septal and lateral e'in separate analyses of the HCM and control groups.**

| Dependent variable | Independent variable | Univariate r | Multivariate β | P value in multivariate model | Cumulative adjusted r² |
|---|---|---|---|---|---|
| **HCM subjects** | | | | | |
| Septal e' | Septal EDExc | 0.66 | 1.0 | <0.001 | 0.40 |
| | Septal EDDur | -0.24 | -0.74 | <0.001 | 0.83 |
| Lateral e' | Lateral EDExc | 0.66 | 1.05 | <0.001 | 0.41 |
| | Lateral EDDur | -0.23 | -0.76 | <0.001 | 0.85 |
| **Control subjects** | | | | | |
| Septal e' | Septal EDExc | 0.82 | 0.84 | <0.001 | 0.65 |
| | Septal EDDur | -0.28 | -0.33 | 0.006 | 0.75 |
| Lateral e' | Lateral EDExc | 0.80 | 0.88 | <0.001 | 0.63 |
| | Lateral EDDur | -0.02 | -0.27 | 0.047 | 0.68 |

and AExc, and therefore any reduction in SExc must also be accompanied by a reduction in the sum of EDExc and AExc. A decrease in EDExc could potentially occur without a decrease in SExc, but it would need to be accompanied by an increase in AExc of similar magnitude to the reduction in EDExc. On the other hand, if a reduction in SExc occurred in the absence of a reduction in EDExc, it would have to be accompanied by a reduction in AExc. During aging between 20 and 80 years there is a reduction in both SExc and EDExc, with an accompanying (but only partly compensatory) increase in AExc [18]. The findings of the present study demonstrate that there are differences in HCM effects on long-axis function compared to those of aging. Although SExc and EDExc decrease in both circumstances, in HCM SExc and EDExc decrease by a similar magnitude and AExc does not increase, whereas during aging EDExc decreases to a greater extent than SExc, and AExc increases. Indeed, not only was there no left atrial-mediated compensation for the reduction of long-axis contraction in HCM, but for the septal wall there was actually a small reduction in AExc. Further highlighting the existence of a direct relationship between SExc and EDExc, in the combined analysis of HCM and control subjects, SExc was the major determinant of EDExc for both the septal and lateral walls. Moreover, variation in SExc explained all the effect of HCM on EDExc as a HCM diagnosis was not an independent predictor of EDExc in models of septal and lateral EDExc which included the respective SExc. In other words, all the reduction of EDExc due to the effects of HCM was explained by the concomitant reduction in LV long-axis contraction.

In studies of aging-effects on early diastolic long-axis motion, a close correlation between e'and EDExc has been evident, with r values between 0.73 and 0.86 [18, 20], however, to our knowledge, the relationship between e'and EDExc has not previously been investigated in HCM. In the control group in the present study, r values for the correlation of e'with EDExc were 0.80–0.82, whereas the r value for both septal and lateral walls in the HCM group was only 0.66. An important difference between the effects of aging and HCM on early diastolic motion was demonstrated in the present study as EDDur was prolonged in HCM, whereas no consistent effect of aging on EDDur has been demonstrated [18]. The contribution of the prolongation of EDDur to the reduction in e'in HCM was investigated in the present study using multivariate analyses, where EDDur and EDExc were entered as independent variables. In the HCM group, EDExc explained only 40–41% of the variances in e', but this increased to 83–85% with the addition of EDDur, with both variables independent contributors in the models. Therefore, while the major effect of aging on e'appears to be related to its effect on EDExc, the effect of HCM on e'can be attributed nearly equally to the reduction in EDExc and the prolongation of EDDur.

On the basis that correlations have been reported between e'and invasive measures of LV relaxation such as tau [23–26], e'has been considered to be a marker of the speed of LV relaxation, and some investigators have assumed that a decrease in e'reflects impairment of the active phase of relaxation at the level of the cardiomyocyte [27, 28]. However, the closeness of the relationship between e'and tau has varied widely [25, 26], and in invasive studies there has not been contemporaneous evaluation of the relation of e'with EDExc or SExc. More recently, it has been demonstrated in a number of studies that there is a correlation between TDI markers of systole and early diastole [14, 15, 18, 29, 30]. Furthermore, in a study of subjects with HCM by Popovic et al, a close correlation of e'with SExc was reported (r = 0.79), and in that study there was no additional contribution to the prediction of e'from a non-invasively estimated value of tau when SExc was included in the model [15]. Indeed, as tau is a calculated variable based on measurements of LV pressure prior to mitral valve opening, and the early diastolic TDI signal does not even begin until mitral valve opening, it has been pointed out previously that e'could only ever be indirectly related to tau [31]. We have extended the findings of Popovic et al in the present study by demonstrating that EDExc is also reduced in HCM, that the

main determinant of the reduction in EDExc is the decrease in long-axis contraction, whereas e'is reduced in HCM via similar contributions from the associated reductions in SExc and EDExc, and the accompanying prolongation of EDDur.

That there was a substantial contribution from prolongation of EDDur to the reduction in e'in HCM is consistent with the possibility that slowing of active relaxation at the level of the cardiomyocyte could be a contributor to the lower e'. IVRT was also prolonged in HCM and this was despite the possibility that a higher left atrial pressure in HCM subjects might have caused a shortening of the IVRT [31]. However, prolongation of contraction duration was also demonstrated in HCM subjects in the present study, there was non-uniformity of contraction between the septal and lateral walls, and non-uniformity of both contraction and relaxation have been previously reported in HCM [32]. Therefore, non-uniformity of contraction and relaxation within a LV wall could both be contributors to the prolongation of EDDur in HCM.

That SExc and s'were both reduced in HCM subjects in the present study is consistent with the findings of previous studies [3–7], although the relationship of s'with SExc has not been the subject of previous investigation in HCM, and it may have been previously assumed that s'and SExc provide similar, and thus redundant, information about long-axis contraction. However, that there are differences in the information provided by s'and SExc about LV long-axis contraction was evident in the combined group as only 49–55% of the variances in s'were explained by the respective SExc. Similar to a previous report in other subject groups [33], heart rate was an independent predictor of s'when combined with SExc in the combined analysis of the HCM and control subjects in the present study. Thus, not only can s'vary independently of SExc, but this variation may well be larger in the setting of a higher or lower heart rate. However, heart rate made only a minor contribution to the prediction of s'as even with the inclusion of heart rate with SExc only 56–59% of the variances in s'could be explained. Although both s'and SExc were lower in HCM, the lack of contribution of a HCM diagnosis in the model of s'which included SExc and heart rate suggests that the relationships of s'with SExc and heart rate were not affected independently by the HCM phenotype.

The increases in A, AExc and a'seen during aging can been attributed to a decrease in early diastolic blood flow into the left ventricle, a resulting increase in blood in the left atrium at the time of its contraction, and then increased left atrial contraction via activation of the Frank-Starling mechanism [34]. The cause of the absence of an increase in AExc in HCM in our study in circumstances where increased atrial stretch was expected can only be speculated on from the available data, as changes in atrial preload, afterload and contractility could all be playing a role.

There are a number of limitations of this study, although it is unlikely that they have had a substantial influence on the main study findings. The subjects were identified retrospectively, however, the echocardiographic studies were performed on the same machine and using a standard protocol, and measurements were performed prospectively using the same software. The study group was small, but it was large enough to enable detection of the expected and postulated differences between HCM and control subjects, and also large enough to show significant correlations on multivariate analysis. HCM is a disease with a large variation in phenotypic expression, whereas the HCM group in this study all had asymmetric LV hypertrophy and most had either resting or inducible LV outflow tract obstruction. It cannot be assumed that our findings apply to other HCM phenotypes. On the other hand, reduction in e'in HCM has been previously demonstrated to be present in subjects with or without LV outflow tract obstruction [35]. A residual beta blocker effect in the HCM subjects could not be excluded, but there was no difference in heart rate between HCM and control groups, and the magnitudes of the septal and lateral e'in HCM subjects were in keeping with previous HCM studies [3, 5].

Measurement of e'in subjects with confirmed HCM may have prognostic utility as a lower e'has been reported to be a predictor of the development of heart failure [36]. However, if this information is to be used in the monitoring of future endeavours to modify the disease course in HCM it is also important to have a full understanding of the mechanism(s) underlying the reduction of e'. In the present study, it was confirmed that a reduction in long-axis contraction is a major contributor to the reductions in both EDExc and e'which occur with HCM, and also demonstrated that prolongation of EDDur is also a substantial contributor to the reduction in e'. An additional implication of the latter finding is that the mechanisms underlying a reduction in e'should not be assumed to be the same in different cardiac conditions. Further investigation will be required to understand the cause of the prolonged EDDur in HCM. A role has been proposed for the measurement of e'during the assessment of subjects with possible HCM based on previous reports that e'is a sensitive test for the early diagnosis of HCM [3, 4, 11]. However, whether changes in EDExc and EDDur are also both contributors to the reduced e'in the early stages of HCM (as we have found them to be in established HCM), cannot be assumed based on our findings, and will require further study.

## Supporting information

**S1 File. Hypertrophic cardiomyopathy long-axis function.**
(XLSX)

## Author Contributions

**Conceptualization:** Wai-ee Thai.

**Data curation:** Roger E. Peverill, Bon Chou, Lesley Donelan, Wai-ee Thai.

**Formal analysis:** Roger E. Peverill, Bon Chou.

**Investigation:** Roger E. Peverill, Bon Chou, Wai-ee Thai.

**Methodology:** Roger E. Peverill, Lesley Donelan, Wai-ee Thai.

**Project administration:** Roger E. Peverill, Lesley Donelan, Wai-ee Thai.

**Resources:** Roger E. Peverill.

**Software:** Roger E. Peverill, Lesley Donelan.

**Supervision:** Roger E. Peverill.

**Validation:** Roger E. Peverill.

**Writing – original draft:** Roger E. Peverill.

**Writing – review & editing:** Roger E. Peverill, Bon Chou, Lesley Donelan, Wai-ee Thai.

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
