## [Decision Letter · Decision Letter 0]

6 Jul 2020

PONE-D-20-18821

Left ventricular long-axis function in hypertrophic cardiomyopathy - Relationships between e`, early diastolic excursion and duration, and systolic excursion

PLOS ONE

Dear Dr. Peverill,

Thank you for submitting your manuscript to PLOS ONE. After careful consideration, we feel that it has merit but does not fully meet PLOS ONE’s publication criteria as it currently stands. Therefore, we invite you to submit a revised version of the manuscript that addresses the points raised during the review process.

Please address in detail the ethical issues related to this manuscript and remove all surnames, please explain.

We look forward to receiving your revised manuscript.

Kind regards,

Elena Cavarretta, M.D., Ph.D.

Academic Editor

PLOS ONE

Journal Requirements:

Reviewers' comments:

Reviewer's Responses to Questions

**Comments to the Author**

1. Is the manuscript technically sound, and do the data support the conclusions?

Reviewer #1: Yes

Reviewer #2: Partly

2. Has the statistical analysis been performed appropriately and rigorously? 

Reviewer #1: Yes

Reviewer #2: Yes

3. Have the authors made all data underlying the findings in their manuscript fully available?

Reviewer #1: Yes

Reviewer #2: Yes

4. Is the manuscript presented in an intelligible fashion and written in standard English?

Reviewer #1: Yes

Reviewer #2: Yes

5. Review Comments to the Author

Reviewer #1: The manuscript by Peverill et al represents a case-control study with the purpose to highlight the differences of a wide range of LV TDI indices in HCM conpared to healthy volunteers. Overall, it's a well written manuscript which analytically sheds some light to the interrelations of the TDI parameters that reflect the LV long axis function in this common cardiomyopathy. In my opinion, however, there are some concerns.

1)Although the statistics do not seem compromised, the study sample is too small even for a rare cardiomyopathy. Moreover, all HCM patients had asymmetrical septal hypertrophy, which is the most common type of HCM, but it does not represent the broad spectrum of phenotypic diversity that characterizes the cardiomyopathy. To lend more support only 2 patients (9%) had non-obstructive phenotype and this is not the case in HCM's general population. Therefore, the results of the study cannot be extrapolated to the general HCM population. This limitation should be clearly stated.

2) The discussion section is too long to maintain the reader's interest. I understand that the results are too many, however, the authors should highlight the most important which may have clinical implications.

3) Throughout the manuscript there are long sentences which loose their meaning. The authors should revise these throughout the manuscript.

4) The most important concern, however, is that the authors state that the need for informed consent was waived as all analyses were performed on anonymized data. However, in the data supporting document at the second sheet and second column there are surnames. Could you explain this disrepancy? If these are the surnames of patients the lack of consents raises serious ethical concerns. In any case these surnames should be removed.

Reviewer #2: This is interesting data reframing old and offering new insights into hypertrophic cardiomyopathy. The Background in the Abstract talks about e' being a measure of LV dysfunction - but shouldn't that be "diastolic" dysfunction? It is known that LVET is prolonged with high pressure gradients in HOCM and I am surprised they did not find a relationship between SDur and the gradient. Or did they? Was there any difference between those with significant gradients vs those without? How did the extent of hypertrophy within the HCM patients affect the data? Did basal hypertrophy have less abnormalities than those with marked hypertrophy? In that vein, the first sentence mentions that HCM is a "hereditary heart disease of dominant inheritance". Actually only about a third of patients have a dominant hereditary background. That should be corrected. It would have been nice to look at E/e' and other parameters as this is used clinically. IVRT shortens with increased left atrial pressure and that should have been mentioned. I am not surprised that e' and EDExc are related because they are different ways of measuring the same "wave". It would have been useful, again for clinical purposes, to look at pulmonary vein pulsed Doppler profile and see how that correlates. On a personal note, I think the discussion is way too long and should be condensed and eliminate most of the speculation.

6. PLOS authors have the option to publish the peer review history of their article (what does this mean?). If published, this will include your full peer review and any attached files.

Reviewer #1: **Yes: **Thomas Zegkos

Reviewer #2: **Yes: **Charles Pollick

---

## [Author Response · Author response to Decision Letter 0]

22 Aug 2020

RESPONSE TO REVIEWERS

We thank the reviewers for their constructive suggestions and have addressed the issues they have raised raised below.

Reviewer #1: The manuscript by Peverill et al represents a case-control study with the purpose to highlight the differences of a wide range of LV TDI indices in HCM conpared to healthy volunteers. Overall, it's a well written manuscript which analytically sheds some light to the interrelations of the TDI parameters that reflect the LV long axis function in this common cardiomyopathy. In my opinion, however, there are some concerns.

1) Although the statistics do not seem compromised, the study sample is too small even for a rare cardiomyopathy. Moreover, all HCM patients had asymmetrical septal hypertrophy, which is the most common type of HCM, but it does not represent the broad spectrum of phenotypic diversity that characterizes the cardiomyopathy. To lend more support only 2 patients (9%) had non-obstructive phenotype and this is not the case in HCM's general population. Therefore, the results of the study cannot be extrapolated to the general HCM population. This limitation should be clearly stated.

# Although the present study can be considered small, it is important that it was already well known that there are substantial differences in LV long-axis TDI findings, and in particular e`, between HCM and age and sex matched control subjects. Large numbers were therefore not going to be required to compare long-axis LV function in HCM subjects with a well-matched control group, and indeed the size of the two groups in the present study proved sufficient to test the study hypotheses. There are therefore no concerns about type 2 error with respect to the main aims of the study. We agree that our findings can only be said to apply to the most common HCM phenotype of asymmetrical septal hypertrophy and there are now additional statements in the limitations paragraph in the discussion with respect to this. Included now are statements about the large variability in the HCM phenotype, the specific phenotypic features of our subject group, and that our results cannot be assumed to reflect long-axis function in subjects with other HCM variants.

2) The discussion section is too long to maintain the reader's interest. I understand that the results are too many, however, the authors should highlight the most important which may have clinical implications.

# A number of less important sections in the Discussion have been removed.

3) Throughout the manuscript there are long sentences which lose their meaning. The authors should revise these throughout the manuscript.

# A number of the longer sentences have been changed.

4) The most important concern, however, is that the authors state that the need for informed consent was waived as all analyses were performed on anonymized data. However, in the data supporting document at the second sheet and second column there are surnames. Could you explain this discrepancy? If these are the surnames of patients the lack of consents raises serious ethical concerns. In any case these surnames should be removed.

# We agree that this was a most concerning oversight. The spreadsheet has been edited to remove the worksheet with identifying data. 

Reviewer #2: This is interesting data reframing old and offering new insights into hypertrophic cardiomyopathy.

The Background in the Abstract talks about e' being a measure of LV dysfunction - but shouldn't that be "diastolic" dysfunction?

# We agree that e` has been commonly thought of as a measure of diastolic function. However, given increasing evidence that e` is determined in part by the previous contraction (see supporting references in manuscript), and that one of the main aims of the present study was to investigate the determinants of e` and EDExc in HCM, the more general term of dysfunction seemed appropriate.

It is known that LVET is prolonged with high pressure gradients in HOCM and I am surprised they did not find a relationship between SDur and the gradient. Or did they? Was there any difference between those with significant gradients vs those without? 

# HCM was associated with prolongation of lateral SDur, but not septal SDur, in the present study. There were no differences in either septal or lateral SDur based on comparison of those HCM subjects with a resting versus those without a resting LV outflow tract gradient >30 mmHg. However, this analysis has not been included in the Results as it was not the focus of the current study, and we also had concern about the possibility of a type 2 error. A previous report of a correlation of LVET (corrected using the square root of the heart rate) with LV-aortic pressure gradient was based on simultaneous invasive measurements of gradient and LVET (Wigle et al Circulation 1967). It is therefore possible that simultaneous gradient and TDI acquisition may be necessary to accurately detect a relationship of the gradient with SDur. 

How did the extent of hypertrophy within the HCM patients affect the data? Did basal hypertrophy have less abnormalities than those with marked hypertrophy?

# This interesting question has been considered in several ways, but because of concern about the possibility of a false negative finding we have not included all of the analyses in the Results, and have not mentioned this issue in the Discussion. The percentage differences in the average e`, EDExc and EDDur between the HCM and control subjects for both septal and lateral walls are now included in the Results. Although the percentage differences in e` and EDExc between the groups were not identical for the two walls (40% v 55% lower for septal and lateral e`, and 25% v 33% lower for septal and lateral EDExc, respectively), neither were they substantially different. The following regression analyses were also performed but have not been included in the manuscript. In HCM subjects there were correlations between septal and lateral e` (r=0.81, p<0.001), septal and lateral EDExc (r=0.66, p=0.001), and septal and lateral EDDur (r=0.64, p=0.001). There were no correlations of septal wall thickness with septal e`, EDExc or EDDur, with or without adjustment for the relevant lateral wall value of these variables (p>0.1 for all). Together the above findings are consistent with an effect of HCM on the long-axis function of both LV walls which is not dependent on variability in wall thickness, and they do not support an additional effect on long-axis function from the degree of septal wall thickening. However, neither do we feel justified in saying we can exclude such an effect. 

Our findings with regard to septal and lateral e` can be compared to those of the cited article by Nagueh et al (Circulation, 2001). Their HCM group had similar septal and posterior wall thickness to our HCM group. Their results showed similar percentage differences between HCM and control subjects for the average e` of the septal and lateral walls (63% lower septal e` and 65% lower lateral e` in HCM subjects), and thus also did not support a major effect of variability in asymmetric septal wall thickness on LV long-axis function.

In that vein, the first sentence mentions that HCM is a "hereditary heart disease of dominant inheritance". Actually only about a third of patients have a dominant hereditary background. That should be corrected.

#The statement regarding dominant inheritance has been removed.

It would have been nice to look at E/e' and other parameters as this is used clinically.

# E/septal e`, E/lateral e` and E/average e` have now been included in Table 2. None of these variables were normally distributed and they have been presented as median [range]and analysis was performed with a Mann-Whitney test. A sentence about the increased magnitude of these variables in HCM subjects is now included in the text of the Results. In the Methods, a reference is cited for why E/e` cannot be used to accurately estimate mean left atrial pressure in HCM. 

IVRT shortens with increased left atrial pressure and that should have been mentioned. 

# A sentence has been added to what is now the 5th paragraph in the discussion regarding the possibility of the effects of left atrial pressure on IVRT with a relevant citation. 

I am not surprised that e' and EDExc are related because they are different ways of measuring the same "wave". 

# As the reviewer suggests, a correlation between e` and EDExc is not surprising, and it was also expected based on previous findings in healthy adult subjects. On the other hand, to our knowledge EDExc had not been specifically investigated in HCM, the relationship of e` with EDExc in HCM has not been investigated, and one of the main aims of this study was to determine if e` in HCM might also be influenced by prolongation of EDDur. A new finding of this study was that prolongation of EDDur did occur in HCM and was an important determinant of the lower e` in HCM in conjunction with the lower EDExc in HCM.

It would have been useful, again for clinical purposes, to look at pulmonary vein pulsed Doppler profile and see how that correlates. 

# Pulmonary vein flow assessment was not routinely performed in the cohort.

On a personal note, I think the discussion is way too long and should be condensed and eliminate most of the speculation.

# The discussion has been reduced in length.

---

## [Decision Letter · Decision Letter 1]

3 Sep 2020

PONE-D-20-18821R1

Left ventricular long-axis function in hypertrophic cardiomyopathy - Relationships between e`, early diastolic excursion and duration, and systolic excursion

PLOS ONE

Dear Dr. Peverill,

Thank you for submitting your manuscript to PLOS ONE. After careful consideration, we feel that it has merit but does not fully meet PLOS ONE’s publication criteria as it currently stands. Therefore, we invite you to submit a revised version of the manuscript that addresses the points raised during the review process.

Please address the minor concerns raised by the reviewer #2.

We look forward to receiving your revised manuscript.

Kind regards,

Elena Cavarretta, M.D., Ph.D.

Academic Editor

PLOS ONE

Reviewers' comments:

Reviewer's Responses to Questions

**Comments to the Author**

1. If the authors have adequately addressed your comments raised in a previous round of review and you feel that this manuscript is now acceptable for publication, you may indicate that here to bypass the “Comments to the Author” section, enter your conflict of interest statement in the “Confidential to Editor” section, and submit your "Accept" recommendation.

Reviewer #1: All comments have been addressed

Reviewer #2: (No Response)

2. Is the manuscript technically sound, and do the data support the conclusions?

Reviewer #1: Yes

Reviewer #2: Partly

3. Has the statistical analysis been performed appropriately and rigorously? 

Reviewer #1: Yes

Reviewer #2: Yes

4. Have the authors made all data underlying the findings in their manuscript fully available?

Reviewer #1: Yes

Reviewer #2: Yes

5. Is the manuscript presented in an intelligible fashion and written in standard English?

Reviewer #1: Yes

Reviewer #2: Yes

6. Review Comments to the Author

Reviewer #1: The manuscript was improved substantially. All my comments were addressed sufficiently and the paper is now publishable.

Reviewer #2: I like the article. It makes one think about the relationship between systolic and diastolic excursion. But I still have the following minor concerns about the revised version.

1. Abstract - "(e') is recognized to be a measure of left ventricular (LV) dysfunction.." Actually it is recognized to be a measure of diastolic dysfunction by most people. While this paper attempts to challenge this notion, for most people this is not considered accurate. I am not content this was not changed or acknowledged.

2. Introduction - first sentence - says "..is a hereditary heart disease.." - this is only true about 35% of the time. I am not content this was not changed to reflect this fact.

3. Table 4 makes no sense. Whereas Table 5 lists the dependent and independent variables correctly, this is not stated for Table 4. HCM per se is not an independent variable.

7. PLOS authors have the option to publish the peer review history of their article (what does this mean?). If published, this will include your full peer review and any attached files.

Reviewer #1: **Yes: **Thomas Zegkos, MD

Reviewer #2: **Yes: **Charles Pollick

---

## [Author Response · Author response to Decision Letter 1]

21 Sep 2020

RESPONSE TO REVIEWERS

We thank the reviewers for their further comments and are very pleased to see hear that Reviewer 1 believes it is satisfactory in its current form. The issues raised by Reviewer 2 are addressed below.

Reviewer #1: The manuscript was improved substantially. All my comments were addressed sufficiently and the paper is now publishable.

Reviewer #2: I like the article. It makes one think about the relationship between systolic and diastolic excursion. But I still have the following minor concerns about the revised version.

# 1. Abstract - "(e') is recognized to be a measure of left ventricular (LV) dysfunction.." Actually it is recognized to be a measure of diastolic dysfunction by most people. While this paper attempts to challenge this notion, for most people this is not considered accurate. I am not content this was not changed or acknowledged.

The authors are concerned with the use of "recognized" together with e` and diastolic dysfunction in that sentence as the word "recognized" implies acknowledgment of the validity of something which is being challenged in the present study. In an attempt to address the reviewer's issue but also avoid the use of "recognized" the first sentence of the abstract has been changed to "The peak velocity of early diastolic mitral annular motion (e`) is believed to provide sensitive detection of left ventricular (LV) diastolic dysfunction in hypertrophic cardiomyopathy..."

# 2. Introduction - first sentence - says "..is a hereditary heart disease.." - this is only true about 35% of the time. I am not content this was not changed to reflect this fact.

The authors agree that "hereditary heart disease" can be considered inappropriate as it implies that it has been inherited in all cases. "Hereditary" has been changed to the more accurate term of "heritable", which is the term that was used in the cited reference from Ommen et al. 

# 3. Table 4 makes no sense. Whereas Table 5 lists the dependent and independent variables correctly, this is not stated for Table 4. HCM per se is not an independent variable.

The authors do not accept that HCM cannot be described as an independent variable in the manner it has been used in the analysis shown in Table 4. The methods used in the analysis are described in the Statistics session in the following sentences. "Multivariate linear regression analysis in the combined HCM and control group was performed to assess the relationships between long-axis systolic and diastolic variables. HCM versus control was included as a dummy variable (1 v 0) as a last step in multivariate models to see if it accounted for further variance in the dependent variable after inclusion of the other independent variables." Testing of the contribution of a categorical variable by including it as a dummy variable in multivariate linear regression analysis is a recognized technique (see section on Categorical Predictor Variables in review by Slinker et al. Multiple linear regression: Accounting for multiple simultaneous determinants of a continuous dependent variable. Circulation 2008; 117: 1732-1737). If there is ongoing concern about this issue then the authors would be pleased for the editor to obtain independent statistical advice.

---

## [Decision Letter · Decision Letter 2]

24 Sep 2020

Left ventricular long-axis function in hypertrophic cardiomyopathy - Relationships between e`, early diastolic excursion and duration, and systolic excursion

PONE-D-20-18821R2

Dear Dr. Peverill,

We’re pleased to inform you that your manuscript has been judged scientifically suitable for publication and will be formally accepted for publication once it meets all outstanding technical requirements.

Kind regards,

Elena Cavarretta, M.D., Ph.D.

Academic Editor

PLOS ONE

Additional Editor Comments (optional):

Reviewers' comments:

Reviewer's Responses to Questions

**Comments to the Author**

1. If the authors have adequately addressed your comments raised in a previous round of review and you feel that this manuscript is now acceptable for publication, you may indicate that here to bypass the “Comments to the Author” section, enter your conflict of interest statement in the “Confidential to Editor” section, and submit your "Accept" recommendation.

Reviewer #2: All comments have been addressed

2. Is the manuscript technically sound, and do the data support the conclusions?

Reviewer #2: Yes

3. Has the statistical analysis been performed appropriately and rigorously? 

Reviewer #2: Yes

4. Have the authors made all data underlying the findings in their manuscript fully available?

Reviewer #2: Yes

5. Is the manuscript presented in an intelligible fashion and written in standard English?

Reviewer #2: Yes

6. Review Comments to the Author

Reviewer #2: I don't want to nitpick about my concerns and will let them go. To say a disease is heritable when most cases are not seems, however, confusing to me. Table 4 also is confusing to me but I accept their statistical explanation, though confess I don't understand it. I expect other readers will also find it confusing. I am pleased they addressed my issue regarding e'

7. PLOS authors have the option to publish the peer review history of their article (what does this mean?). If published, this will include your full peer review and any attached files.

Reviewer #2: **Yes: **Charles Pollick

---

## [Editor Report · Acceptance letter]

28 Sep 2020

PONE-D-20-18821R2 

Left ventricular long-axis function in hypertrophic cardiomyopathy - Relationships between e`, early diastolic excursion and duration, and systolic excursion 

Dear Dr. Peverill:

I'm pleased to inform you that your manuscript has been deemed suitable for publication in PLOS ONE. Congratulations! Your manuscript is now with our production department. 

Kind regards, 

on behalf of

Dr. Elena Cavarretta 

Academic Editor

PLOS ONE